# Identification of Potential *Leishmania* N-Myristoyltransferase Inhibitors from *Withania somnifera* (L.) Dunal: A Molecular Docking and Molecular Dynamics Investigation

**DOI:** 10.3390/metabo13010093

**Published:** 2023-01-06

**Authors:** Mohamed A. A. Orabi, Mohammed Merae Alshahrani, Ahmed M. Sayed, Mohamed E. Abouelela, Khaled A. Shaaban, El-Shaymaa Abdel-Sattar

**Affiliations:** 1Department of Pharmacognosy, College of Pharmacy, Najran University, Najran 61441, Saudi Arabia; 2Department of Clinical Laboratory Sciences, Faculty of Applied Medical Sciences, Najran University, Najran 61441, Saudi Arabia; 3Department of Pharmacognosy, Faculty of Pharmacy, Nahda University, Beni-Suef 62513, Egypt; 4Department of Pharmacognosy, Faculty of Pharmacy, Al-Azhar University, Assiut-Branch, Assiut 71524, Egypt; 5Center for Pharmaceutical Research and Innovation, Department of Pharmaceutical Sciences, College of Pharmacy, University of Kentucky, Lexington, KY 40536, USA; 6Department of Medical Microbiology and Immunology, Faculty of Pharmacy, South Valley University, Qena 83523, Egypt

**Keywords:** leishmaniasis, *L. major*, N-myristoyltransferase, molecular docking, molecular dynamics

## Abstract

Leishmaniasis is a group of infectious diseases caused by *Leishmania* protozoa. The ineffectiveness, high toxicity, and/or parasite resistance of the currently available antileishmanial drugs has created an urgent need for safe and effective leishmaniasis treatment. Currently, the molecular-docking technique is used to predict the proper conformations of small-molecule ligands and the strength of the contact between a protein and a ligand, and the majority of research for the development of new drugs is centered on this type of prediction. *Leishmania* N-myristoyltransferase (NMT) has been shown to be a reliable therapeutic target for investigating new anti-leishmanial molecules through this kind of virtual screening. Natural products provide an incredible source of affordable chemical scaffolds that serve in the development of effective drugs. *Withania somnifera* leaves, roots, and fruits have been shown to contain withanolide and other phytomolecules that are efficient anti-protozoal agents against *Malaria, Trypanosoma*, and *Leishmania* spp. Through a review of previously reported compounds from *W. somnifera*-afforded 35 alkaloid, phenolic, and steroid compounds and 132 withanolides/derivatives, typical of the *Withania* genus. These compounds were subjected to molecular docking screening and molecular dynamics against *L. major* NMT. Calycopteretin-3-rutinoside and withanoside IX showed the highest affinity and binding stability to *L. major* NMT, implying that these compounds could be used as antileishmanial drugs and/or as a scaffold for the design of related parasite NMT inhibitors with markedly enhanced binding affinity.

## 1. Introduction

Leishmaniasis is a serious neglected tropical disease that causes several illnesses connected to immune-system dysfunction and poverty, which are sometimes associated with high fatality rates [1]. The disease is manifested in three different ways: Cutaneous leishmaniasis (CL), which is characterized by lesions at the site of infection; mucocutaneous leishmaniasis, which is characterized by invasion and destruction of the mucosa; and visceral leishmaniasis, which is the most severe form because the infection spreads to other organs like the spleen and liver [2]. Each year, around two million new cases are reported worldwide, with cutaneous leishmaniasis accounting for 75% of these infections. The primary cause of cutaneous leishmaniasis is *Leishmania major,* and out of the 89 countries where it is present, Afghanistan, Brazil, Iran, Peru, Saudi Arabia, and Syria account for 90% of CL cases [1,3]. Despite the long history of the disease, no vaccine is available, and all of the current medications are either ineffective, have unfavorable side effects, or are losing efficacy as resistance emerges [4]. 

To produce a new anti-*Leishmania* drug, long isolation, purification, or synthesis processes, as well as excessively expensive in vitro and in vivo biological evaluations, are required [5]. Thus, the present challenge is to identify novel molecules with potential antileishmanial activity while conserving time and money. Utilizing in silico techniques can shorten the time and lower the cost associated with developing new medications [6]. 

Molecular docking is an in silico technique used to identify correct conformations of small-molecule ligands and estimate the strength of the protein–ligand interaction [7]. Presently, the majority of research for the development of novel drugs focuses on this kind of analysis. Molecular docking has been employed in multiple studies to create new antiparasitic medications while considering a variety of targets to find new *Leishmania* treatments [8]. Discovering new leishmaniasis treatments is made more enticing by molecules implicated in parasite-specific metabolic processes [9]. The affection of this kind of target implies the death of the parasite and the control of the infection [5]. Worth mentioning is that standard chemotherapeutic antileishmanial drugs operate via related mechanisms—namely, interference in parasite metabolic processes (Table 1) [10]. Targeting specific molecular pathways is a common approach in rational drug design and discovery for developing such leishmaniasis-treating compounds. Pteridine reductase, trypanothione reductase, N-myristoyltransferase (NMT), trypanothione synthetase, inosine-uridine nucleoside hydrolase, and topoisomerases are just a few of the more than 21 potential therapeutic targets of antileishmanial drug discovery that have been reported in the literature and gathered in a book chapter [9]. 

N-myristoyltransferase is an example of the molecules implicated in parasite metabolic processes. It is an enzyme that catalyzes the co-translational transfer of a C_14_ fatty acid from myristoyl-CoA onto the N-terminal glycine residue of a significant subset of proteins [11]. N-myristoylation is believed to play a crucial role in the correct cellular localization and biological function of such modified proteins. Since 1997, it has been established that *Leishmania major* contains NMT [12]. Later, it was determined that *Leishmania* sp. needed the NMT enzyme to survive. It has been extensively studied in some organisms, including *Trypanosoma brucei*, *T. cruzi*, and *Leishmania* spp. [13,14,15,16,17]. In a high-throughput screening campaign run by Pfizer, four different scaffolds, aminoacylpyrrolidine (PF-03402623), piperidinylindole (PF-03393842), and thienopyrimidine (PF-003494M), together with a subsequently created fused hybrid compound, 43, were discovered to be highly effective inhibitors of the *Labrus donovani* NMT (Figure 1) [18]. A different experiment on a collection of 1600 pyrazolyl sulfonamide compounds led to the discovery of compound 2, a potent inhibitor of *Leishmania major* NMT with negligible activity against *Leishmania donovani* intracellular amastigotes [5]. Furthermore, several publications and patents indicated the existence of additional NMT inhibitor scaffolds, including pyrrolidines, piperidinylindoles, azetidinopyrimidines, aminomethylindazoles, benzimidazoles, thienopyrimidines, biphenyl derivatives, benzofuranes, benzothiophenes, oxadiazoles, (pyrazolomethyl)-1,3,4-oxadiazoles, and thienopyrimidine [19,20,21,22]. 

N-myristoyltransferase inhibitor compounds from medicinal plants may be useful in the quest for alternatives to current treatments for CL. *Withania somnifera* (L.) Dunal (Solanaceae), also known as Ashwagandha or Indian ginseng, has been used as a traditional herb against a plethora of human medical conditions [23]. It is one of the most extensively used plants in the Unani and Ayurvedic systems of medicine. It has been reported as an important source of withanolides, alkaloids, steroids, flavonoids, nitrogen-containing compounds, and others [23]. Among them, withanolides are a class of highly oxygenated steroids generated from a C28 ergostane skeleton (Figure 2). They are marker compounds characteristic of Solanaceae plants, particularly those of the genus *Withania* [23]. They are thought to be responsible for the majority of *W. somnifera* bioactivity [16]. *W. somnifera* happens to be one of the prime examples of Rasayana, a branch of Ayurvedic science, a medicinal plant that possesses immunomodulation, anti-cancer, anti-depressant, and neuroprotective properties; promotes the body’s resistance to diseases; increases strength and intellect; and delays aging; as well as other biological properties [23]. Closer examination of pertinent in vitro and in vivo studies revealed that *W. somnifera* extracts molecules’ significant bioactivity against several metabolic, reproductive, cardiovascular, neurological, and psychological conditions. Additionally, it has been shown to have antibacterial and antiparasitic properties against *Trypanosoma*, *Leishmania* species, and *Malaria* [24]. The alleged value of *W. somnifera* in treating leishmaniasis, as proposed by earlier research, is highlighted in the following themes. 

### 1.1. Overview of the Antileishmanial Properties of Withania somnifera

Screening of *W. somnifera* from different geographical regions has shown that methanolic extract exhibits in vitro antileishmanial action against free-living promastigotes [25,26] and intracellular amastigotes of *L. major* [26]. Investigation of a solvent-soluble fraction of root and fruit hydromethanolic extracts demonstrated significant antileishmanial promastigote properties of the butanol-soluble fraction from roots and fruits, whereas in vitro growth-inhibitory assessment on axenic amastigotes revealed promising activity of the root ethyl acetate-soluble and butanol-soluble fractions [1]. Further bio-guided fractionation revealed that the withanolide-enriched fraction from *W. somnifera* ethanolic extract was effective [17]. 

Studies have shown that pure withaferin-A, a prominent withanolide in *W. somnifera*, possesses antileishmanial activity. It inhibits protein kinase C, which allows the apoptotic topoisomerase I-DNA complex to induce apoptosis [25,27]. Additionally, withanolide Z, a chlorinated withanolide, has been reported to exert an inhibitory effect against *L. donovani* topoisomerase-1 [28]. 

The anti-leishmanial activity of the withanolide-enriched extract and pure withanolide of *W. somnifera* was found to be mediated by the induction of morphological alterations from a spindle to round shape and the loss of flagella and cell integrity in promastigotes. They also induce apoptosis-like cellular death in *L. donovani* by inducing DNA nicks, apoptosis, and cell-cycle arrest in a dose- and time-dependent manner, actions that were mediated by increasing reactive oxygen species (ROS) production and decreasing mitochondrial potential [29]. Additionally, these withanolide-enriched fractions and pure withanolide of *W. somnifera*, alone or in combination with other herbal products or standard anti-*Leishmanial* drugs, were found to modulate hamsters’ immunological response to infection with *L. donovani*.

### 1.2. Immunomodulatory Effects of W. somnifera in Leishmaniasis Infections

One of the immunopathological consequences of active visceral leishmaniasis is the suppression of protective T-helper (Th)-1 cells and the induction of disease-promoting Th-2 cells [30]. Therefore, host immunomodulation is crucial for the treatment of visceral leishmaniasis. In animal-model research, *W. somnifera* chemotype NMITLI-101 R extract, withaferin A, and the chemotype NMITLI-101 R extract in combination with an ED_50_ dose of miltefosine were investigated for their immunoprotective and therapeutic effects against *L. donovani* infection. It has been observed that the efficacy of *W. somnifera* was linked to the compelling Th1 immune responses driven by interferon-gamma (IFN-γ) and interleukin-12 (IL-12), as well as dramatically reduced levels of Th2 cytokines (IL-4, IL-10) and transforming growth factor beta (TGF-*β*). Meanwhile, they significantly increased the levels of NO production, ROS creation, and *Leishmania*-specific IgG2 antibodies, along with profoundly delayed-type hypersensitivity (DTH) and strong T-cell responses [31,32,33,34].

In another study, when *Asparagus racemosus* and *W. somnifera* were used to treat infected mice, the parasite burden was successfully reduced and protective Th1-type immune responses were elicited, resulting in the normalization of biochemical and hematological parameters [30]. In the treatment of visceral leishmaniasis, the combination of *W. somnifera* extract with cisplatin led to a notable selective elevation of the Th1 type of immunity, verified immunomodulatory action, and a protective impact against the adverse effects of cisplatin on multiple bodily organs. The percentage of CD4 and CD8 T-lymphocytes, as well as the natural killer (NK) cell-associated marker NK1, increased significantly [33].

### 1.3. Molecular Docking Studies in the Development of Antileishmanial Drugs

According to molecular modeling and dynamic investigations of the leishmanial protein kinase C structure, withaferin A and withanone were suggested to disrupt the protein kinase C (PKC) pathway [23,35]. Molecular-docking studies of the binding mode of withaferin-A with pteridine reductase 1 (PTR1), which is involved in pteridine salvage, a crucial enzyme for parasite proliferation, revealed that withaferin-A inhibits PTR-1 enzymes through the uncompetitive mode of inhibition in the parasites. 

In summary, in rational antileishmanial drug design and discovery, targeting particular biochemical pathways was found to be a typical strategy for creating leishmaniasis-treating molecules. In order to screen molecules from both natural and synthetic sources, more than 21 molecular targets were used [9]. The structure-guided creation of new lead compounds discovered in high-throughput screening efforts aimed at *L. major* and *L. donovani* NMT has led to the identification of effective inhibitors [9,36]. These newly discovered *Leishmania* NMT inhibitors did not possess the same cellular activity as the enzyme against *Leishmania donovani* axenic amastigotes, a fact that has been explained by the restricted cellular absorption related to the basic nature of the compounds [19]. 

*W. somnifera’s* wide variety of the non-basic withanolide derivative and phenolics would be an important source of new hits of *Leishmania* NMT inhibitors with higher cellular activity against *Leishmania* parasites. Thus, the goals of this study are to highlight *W. somnifera* anti-leishmanial properties as well as to discover new *L. major* NMT inhibitors using in silico molecular-docking and molecular-dynamics analysis of *W. somnifera* metabolites as potential anti-*L. major* medicine.

## 2. Experimental

### 2.1. Phytochemical Review and Data Collection

A thorough review of the previously reported compounds from *W. somnifera* (L.) Dunal in peer-reviewed research papers and international databases, including Science Direct, Pubchem, Google Scholar, SciFinder, etc., afforded a total of 167 different phytoconstituents, which were considered for the present study.

### 2.2. Molecular-Docking Simulation

PyRx software was used for the docking experiments [37]. The RCSB Protein Data Bank (https://www.rcsb.org/ accessed on 20 March 2022) was used to retrieve the three-dimensional (3D) structure of *L. major* N-myristoyltransferase (NMT) in complex with the thienopyrimidine inhibitor IMP-0000096 (PDB ID: 6QDF) (Figure 3) determined by X-RAY diffraction (Resolution: 1.49 Å) as a target for molecular-docking studies [21]. The 3D chemical structures of the selected molecules were retrieved from PubChem, with polar hydrogen added, partial charge corrected, and energy minimized using the Merck molecular force field (MMFF94x). Molecular-docking analysis was carried out via flexible ligand-fixed receptor-docking parameters using an active complexed ligand active site. The most stable affinity-binding interactions were selected based on the best pose scores. The docking scores and 2D and 3D interactions were recorded [38,39]. BIOVIA Discovery Studio (v21.1.0.20298) was used for 2D and 3D interaction visualization [40].

### 2.3. Molecular-Dynamics Simulation

Molecular-dynamics simulations (MDS) for the best three generated ligand–enzyme complexes were performed using the Nanoscale Molecular Dynamics (NAMD) 3.0 software, applying the CHARMM27 force field, and the MDS was continued for 100 ns following the previously described protocol [41]. The trajectory was stored every 0.1 ns and further analyzed with the VMD 1.9 software. The Molecular Mechanics Poisson–Boltzmann Surface Area (MMPBSA) embedded in the MMPBSA.py module of AMBER18 was utilized to calculate the binding free energy of the docked complex [42]. One hundred frames were processed from the trajectories in total, and the binding free energy was estimated using the following equation:Δ*G*_Binding_ = Δ*G*_Complex_ − Δ*G*_Receptor_ − Δ*G*_Inhibitor_

Each of the aforementioned terms requires the calculation of multiple energy components, including van der Waals energy, electrostatic energy, internal energy from molecular mechanics, and a polar contribution to solvation energy.

## 3. Results

Earlier phytochemical research on *W. somnifera* led to the isolation of many phytochemical compounds. According to their chemical structures and the number of isolates, to the best of our knowledge, it could be distinguished as 10 alkaloids, 15 phenolic compounds, 10 sterols, 6 withanones, 6 chloro-containing withanolides, 5 sulfur-containing withanolides, 9 withanamide, 87 withanolides, and 19 withanosides. To discover potential hits that could be used as scaffolds for developing antileishmanial drug candidates, we evaluated these compounds against the *Leishmania major* NMT utilizing in silico molecular docking and molecular dynamics. The compounds are sorted according to their phytochemical groups parallel to their docking results in descending order according to pose score in Table 2.

## 4. Discussion

Cutaneous leishmaniasis, caused by more than 20 species of *Leishmania* parasites, is a derelict tropical disease endemic in most world nations with high incidence rates. In the Kingdom of Saudi Arabia, CL is a significant public-health issue because of several risk factors, including rapid population growth and migration. The disease is endemic in many parts of the kingdom, and *L. major* and *L. tropica* are the most commonly detected species [123]. Current chemotherapeutic drugs used for leishmaniasis treatment are discouraging due to associated toxicity, a rase of drug resistance, and high cost. The development of new medications with enough safety and affordability is immediately needed to tackle this disease. *W. somnifera* is a perennial shrub found in open fields and deserts from the Mediterranean region to Southeast Asia [1]. A reasonable level of safety has been supported by up to 30 clinical studies; there were no obvious side effects or changes in hematological, biochemical, or vital indicators. Pre-clinical studies on chronic toxicity that lasted up to 8 months [16] further supported the safety. As stated above, both in vitro and in vivo tests were used to determine the antileishmanial activity of various organ extracts and sub-extracts of *W. somnifera*. However, only a small subset of metabolites was tested for antileishmanial activity. Together, these factors make it more important than ever to thoroughly research the phytoconstituents of the plant as possible sustainable and affordable antileishmanial remedies. Due to the economic issue and the hazards of handling a live *Leishmania* parasite in laboratory work, we adopted computational analysis, molecular docking, and molecular dynamics as a quick and cost-effective scheme to identify potential anti-*L. major* compounds from *W. somnifera*.

N-Myristoyltransferase is a prevalent essential enzyme in all *Leishmania* species [21]. NMT catalyzes the binding of myristate to the amino-terminal glycine residue of a subset of eukaryotic proteins involved in various cellular processes, such as vesicular protein trafficking and signal transduction. NMT has been demonstrated to be essential for viability by classical gene knockout and RNA interference, suggesting that this enzyme has the potential as a target for drug development. It has been proven that NMT is a valid therapeutic target for the treatment of fungus and parasite infections [15,124]. The discovery of the NMT crystal structure establishes a method for structural analysis of inhibitor complexes and structure-assisted drug discovery [21]. Amazingly, it has been demonstrated that the inhibitors only operate on the host enzyme despite the high degree of conservation between the active regions of the parasite and human NMTs [125]. According to the determined structure of *β*-sheet in the *C*-terminal domain and protein loops, the residues that are predicted to be the targeted inhibitor site in the NMT are Tyr-80, Val-81, Glu-82, Phe-88, Phe-90, Tyr-92, Asn1-67, Thr-203, Tyr-217, His-219, Phe-232, Tyr-326, Ile-328, Ser-330, Leu-341, Ala-343, Tyr-345, Val-374, Asn-376, Asp-396, Leu-399, Met-420, and Leu421 [126].

As a first step toward utilizing one or more of them as an effective CL medication, the wide range of previously reported compounds from *W. somnifera* species was chosen for virtual screening as *L. major* NMT inhibitors. In total, 167 compounds were studied for their binding affinity as NMT enzyme inhibitors using molecular-docking simulation (Table 2). The results revealed a wide range of compounds’ affinity to the NMT enzyme (−9.0 to −24.0 kcal/mol). 

The compounds with a score of less than −18.0 kcal/mol are thought to be the most active. To evaluate the experimental stability of the docked ligand conformers, these compounds were filtered by RMSD value with a cutoff of 1.7 Å [127]. The top-scoring compounds were calycopteretin-3-rutinoside, withanamide F, withanolide A, (4*β*,5*β*,6*β*,17*α*,22R) 5,6-Epoxy-4,17,27-trihydroxy-1-oxowitha-2,24-dienolide, withanoside II, withanoside X, physagulin D (1→6)-*β*-D-glucopyranosyl-(1→4)-*β*-D-glucopyranoside, withanoside IX, sitoindoside IX, 4,16-dihydroxy-5*β*,6*β*-epoxyphysagulin D, and 24,25-dihydrowithanoside VI. These compounds showed the highest affinity for the NMT active site within the selected parameter range (Figure 4). 4,16-Dihydroxy-5*β*,6*β*-epoxyphysagulin D, calycopteretin 3-rutinoside, and withanoside IX revealed the most potent affinity to the receptor active site with −24.0, −23.3, and −22.2 kcal/mol, respectively.

Targeting with a pose score of −24.0 kcal/mol and an RSMD value of 1.07, 4,16-dihydroxy-5,6-epoxyphysagulin D demonstrated the highest affinity to the selected enzyme. Hydrogen bonds were formed between the compound and Met-420 and Asn-167 as H-donors, as well as interactions with Asn-383, Tyr-80, Val-378, and Asn-167 as H-acceptors (Figure 5). 

The flavonoid glycoside calycopteretin 3-rutinoside was found to have a potential affinity with a pose score value of −23.3 kcal/mol (RSMD = 1.24 Å) with a hydrogen-bond interaction with Met-420 and Leu-421 as H-donor in addition to Tyr-80 as H-acceptor (Figure 6). 

Furthermore, the withanolide glycoside withanoside IX formed hydrogen bonds with Glu-82, Asp-83, Val-346, Val-346, and Met-420, yielding a binding score of −22.2 kcal/mol (RSMD = 1.41; Figure 7). Other hydrophobic interactions were found in all interacted compounds, as shown in Figure 7. 

Compared with the pose score of the complexed thienopyrimidine inhibitor IMP-0000096 (−14.2 kcal/mol), these compounds could be promising antileishmanial scaffolds.

The introduction of promastigote into the body, the blood circulation of amastigote, phagocytosis and macrophage cell growth, lysis, and blood flow as amastigote once more are among the steps that occur in *Leishmania* infection [1]. However, NMT inhibitors obtained by synthesis from various research groups were unable to kill parasites inside macrophages at reasonable therapeutic doses, as shown by in vivo assay [2]. On the other hand, withanolide-enriched extracts from *W. somnifera* reduced the intracellular parasite load by ~50% compared to the infected control [34]. Withaferin-A produced a dose-dependent decrease in parasite number inside the macrophages at concentrations of 0.5–1.5 µM [34]. These literature results, together with our findings from docking studies, suggest that the compounds with the top docking scores could be leishmanicidal agents that operate by inhibiting *L. major* NMT in the free promastigote and the intercellular amastigote forms. Another exciting outcome of our research is the potential to use these molecules as a promising framework for the development of novel leishmanial NMT inhibitors and possibly NMT inhibitors of other protozoa.

### Molecular-Dynamics Simulation Study

To validate the docking results, we subjected the retrieved docking poses to a 100 ns-long molecular-dynamics simulation (MDS) run. As depicted in Figure 8, calycopteretin-3-rutinoside, withanoside IX, and 4,16-dihydroxy-5*β*,6*β*-epoxyphysagulin D achieved good binding stability inside NMT’s active site, showing average RMSDs (i.e., 1.1 Å, 2.2 Å, and 3.9 Å, respectively) that were comparable with that of the co-crystallized inhibitor (i.e., 1.4 Å). The global dynamic behavior of NMT did not show significant change upon binding with either the co-crystallized inhibitor or with the three selected ligands (i.e., calycopteretin-3-rutinoside, withanoside IX, and 4,16-dihydroxy-5*β*,6*β*-epoxyphysagulin D), where the RMSFs of either the ligand-free enzyme or the enzyme bound to the three ligands and the co-crystalized inhibitor showed good alignment (Figure 8B). 

The frequency of H-bonds detected for each ligand along with the co-crystalized inhibitor averaged around two H-bonds (cut-off distance for H-bonds was set to 3.0 Å; Figure 9). In regard to the interaction energies (i.e., electrostatic + van der Waals energies) of each ligand, calycopteretin-3-rutinoside showed the worst average total interaction energy (−26.4 kcal/mol), whereas withanoside IX and 4,16-dihydroxy-5*β*,6*β*-epoxyphysagulin D, along with the co-crystalized inhibitor, showed convergent average total interaction energies (−55.5, −54.5, and −44.4 kcal/mol, respectively; Figure 10).

Accordingly, the calculated binding free energies (Δ*G*_binding_) extracted from the MDS runs were also convergent, ranging from −7.3 kcal/mol for the co-crystallized inhibitor to −9.8 kcal/mol for 4,16-dihydroxy-5,6-epoxyphysagulin D, except for calycopteretin 3-rutinoside, which obtained the lowest negative value (Table 3).

## 5. Conclusions

The urgent need to discover a new anti-*L. major* drug in a reasonable time and at a low cost encouraged us to consider in silico techniques to examine *W. somnifera* metabolites as an inhibitor of *L. major* NMT. Among the 167 virtually screened compounds, the phenolic glycoside calycopteretin-3-rutinoside, withanoside IX, and 4,16-dihydroxy-5*β*,6*β*-epoxyphysagulin D showed promising binding affinity towards *Leishmania* NMT. In light of the fact that the standard chemotherapeutic antileishmanial medications, such as antimonials, amphotericin B, miltefosine, paromomycin, and pentamidine, work by interfering with parasite metabolic processes, exploring new natural substances that are connected to parasite-specific metabolic processes, such as our discovered NMT inhibitors from *W. somnifera*, is extremely promising. The results of our investigation will help scientists design in vitro and preclinical animal studies employing *W. somnifera* metabolites, with an emphasis on the promising NMT inhibitors we have discovered.

## Figures and Tables

**Figure 1 metabolites-13-00093-f001:**
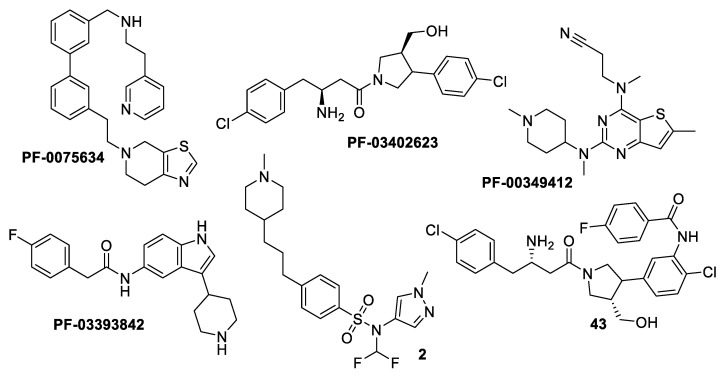
Structures of examples of NMT inhibitors with antileishmanial activity [9].

**Figure 2 metabolites-13-00093-f002:**
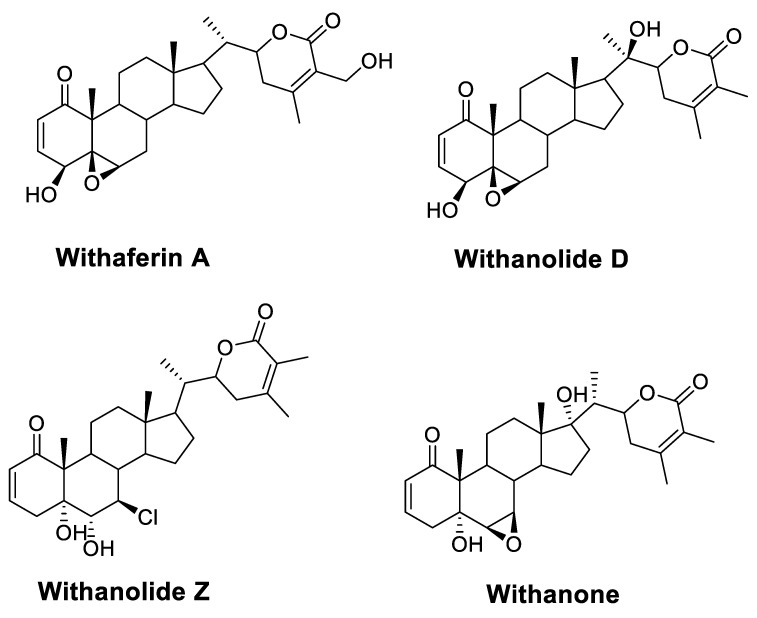
Structural features of bioactive withanolides of *W. somnifera* represented by withaferin A, withanolides D and Z, and withanone.

**Figure 3 metabolites-13-00093-f003:**
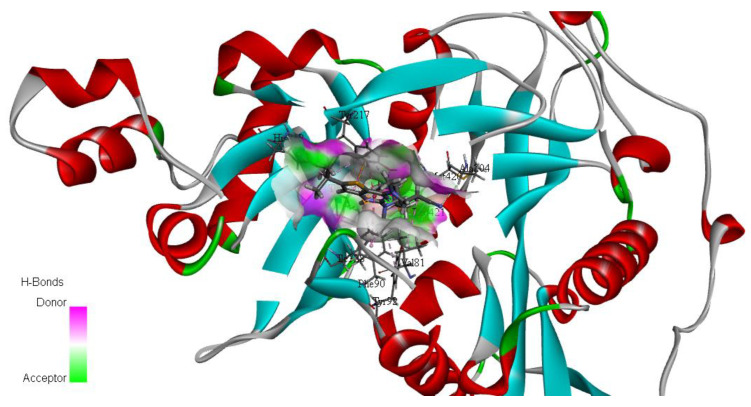
3D structure of *L. major* NMT in complex with the thienopyrimidine inhibitor IMP-0000096 (PDB ID: 6QDF).

**Figure 4 metabolites-13-00093-f004:**
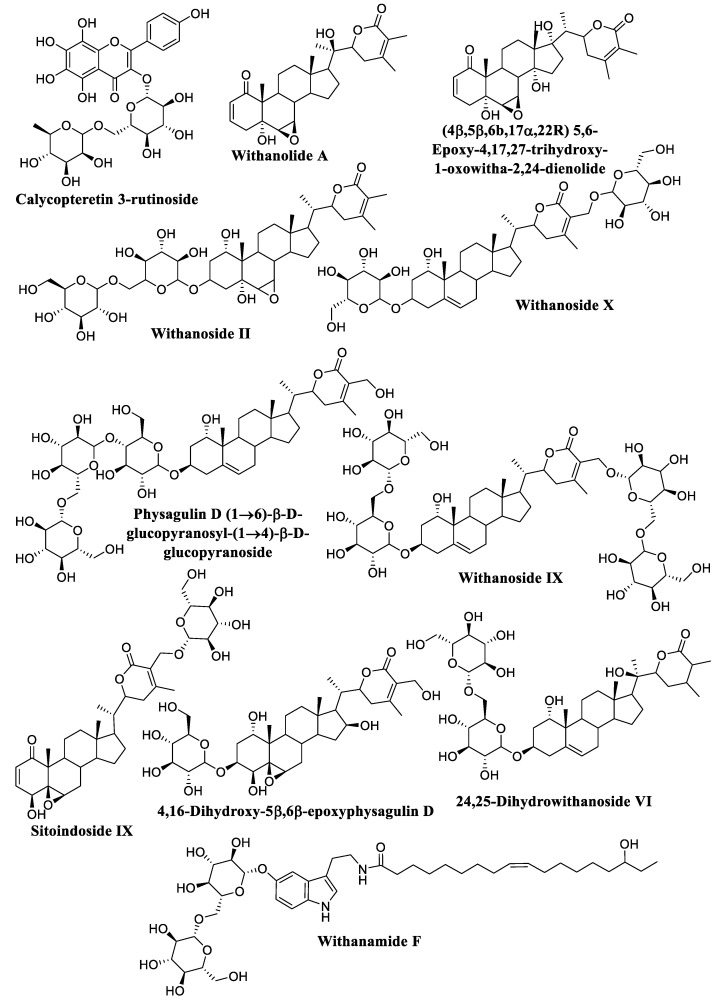
The structures of top-scoring compounds.

**Figure 5 metabolites-13-00093-f005:**
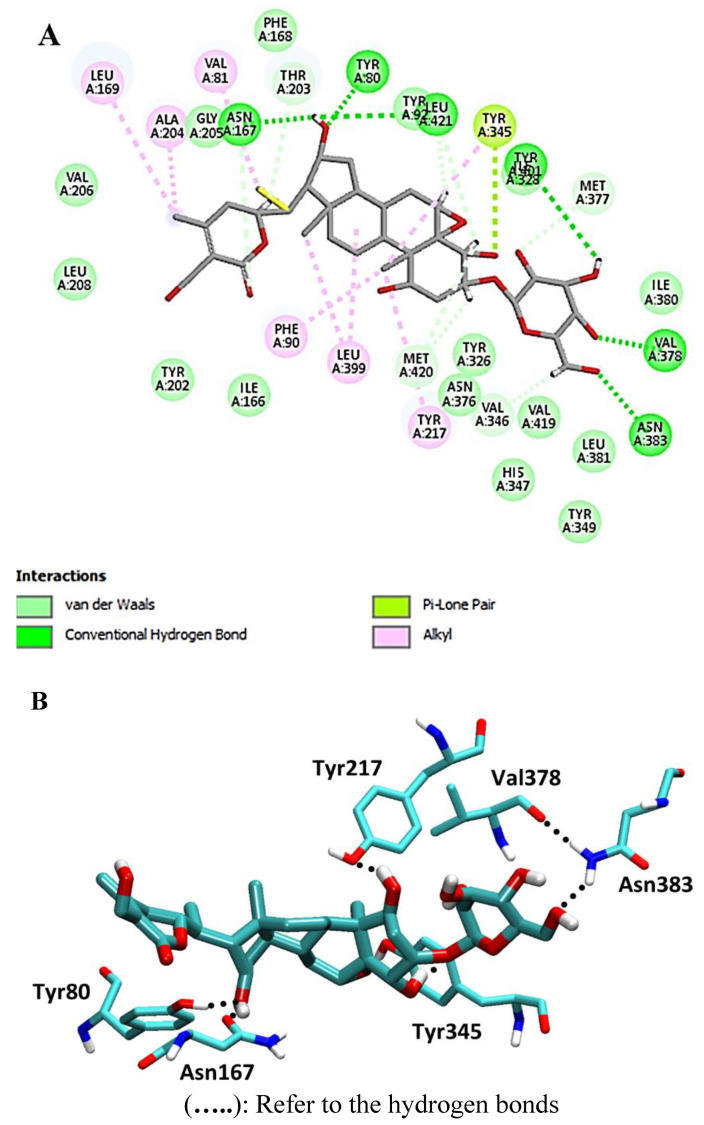
2D and 3D interaction complex of 4,16-dihydroxy-5,6-epoxyphysagulin D with NMT ((**A**) and (**B**), respectively).

**Figure 6 metabolites-13-00093-f006:**
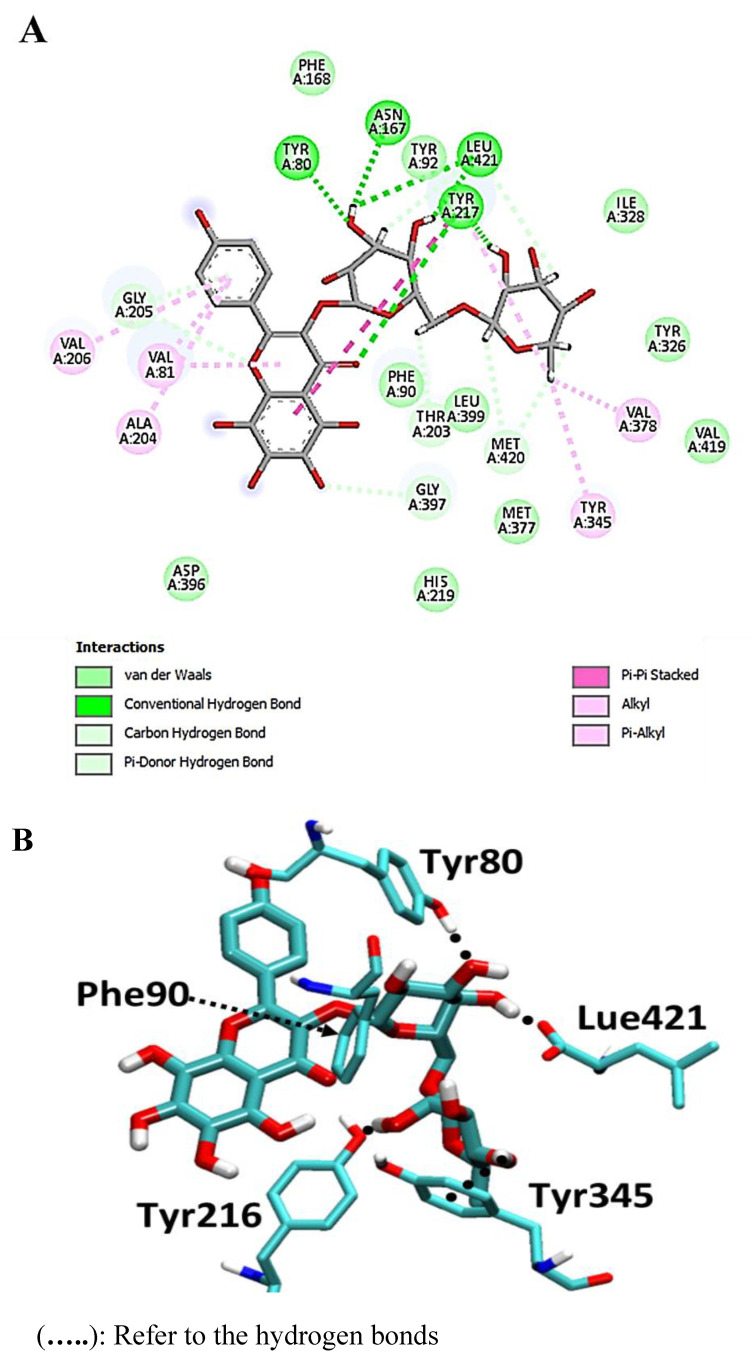
2D and 3D interactions complex of calycopteretin 3-rutinoside with NMT ((**A**) and (**B**), respectively).

**Figure 7 metabolites-13-00093-f007:**
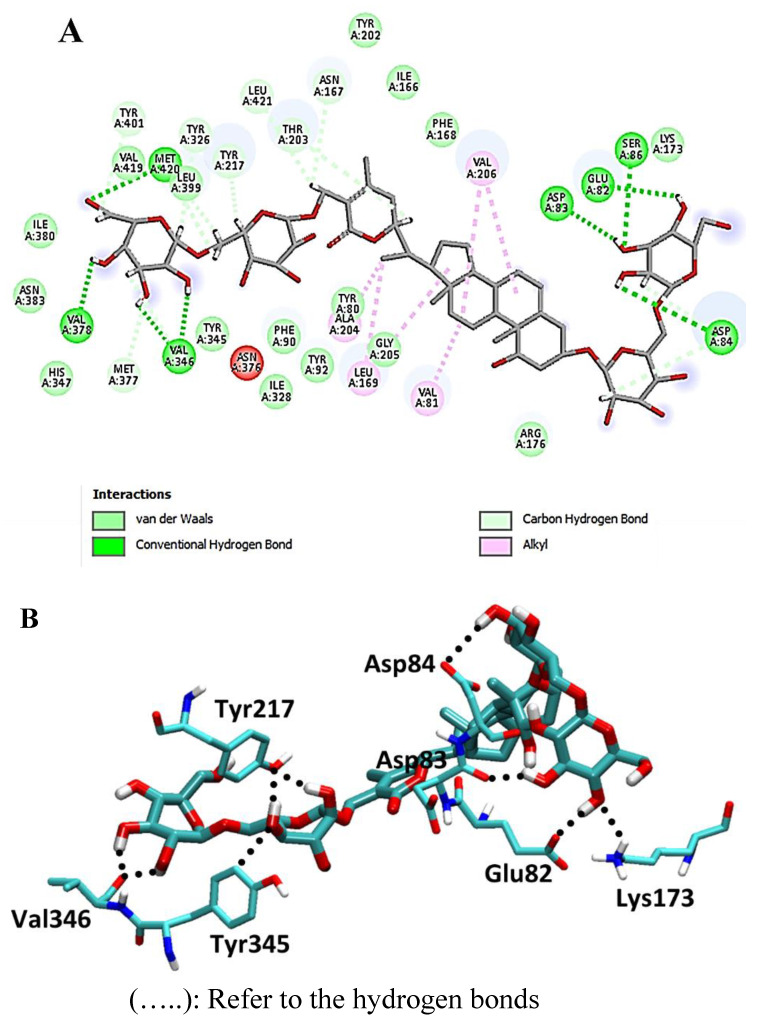
2D and 3D interaction complex of withanoside IX with NMT ((**A**) and (**B**), respectively).

**Figure 8 metabolites-13-00093-f008:**
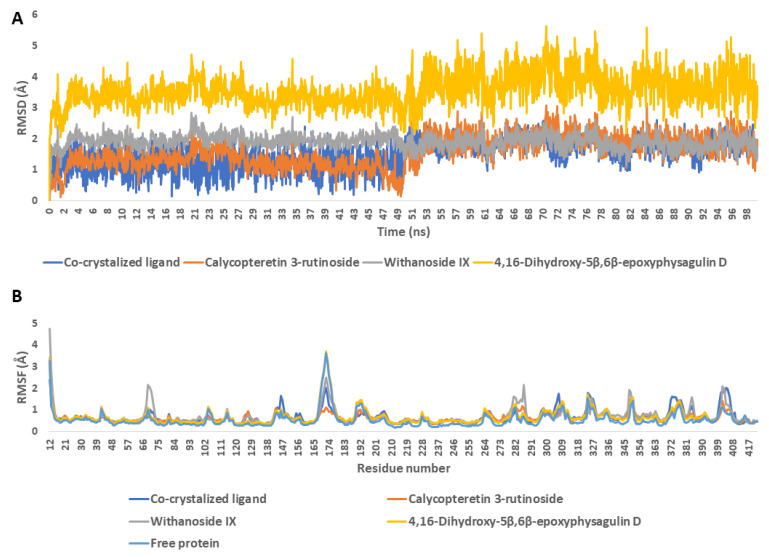
(**A**) RMSDs of the compounds calycopteretin-3-rutinoside, withanoside IX, and 4,16-dihydroxy-5*β*,6*β*-epoxyphysagulin D along with the co-crystalized inhibitor inside the active site of NMT (PDB ID: 6QDF) throughout 100 ns MDS runs. (**B**) RMSFs of NMT in the absence and in the presence of calycopteretin-3-rutinoside, withanoside IX, and 4,16-dihydroxy-5*β*,6*β*-epoxyphysagulin D along with the co-crystalized inhibitor.

**Figure 9 metabolites-13-00093-f009:**
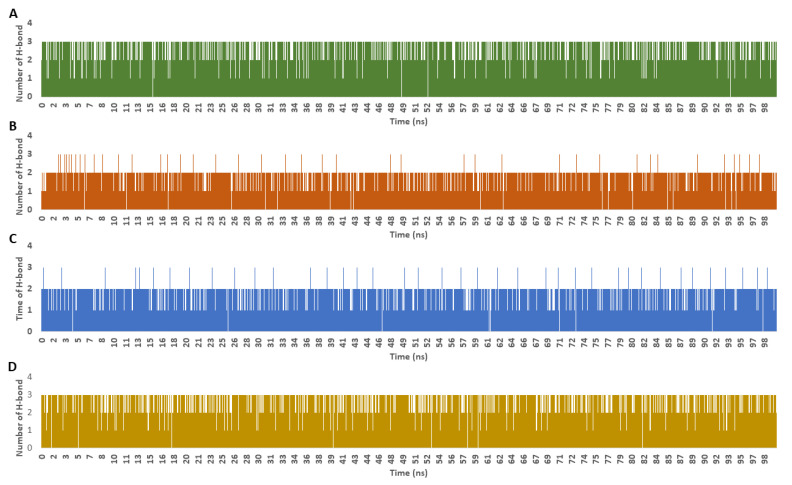
Number of H-bonds detected for calycopteretin-3-rutinoside, withanoside IX, and 4,16-dihydroxy-5*β*,6*β*-epoxyphysagulin D along with the co-crystallized inhibitor inside the active site of NMT throughout 100 ns MDS runs ((**A**–**D**), respectively). The cut-off distance for H-bonds was set to 3.0 Å.

**Figure 10 metabolites-13-00093-f010:**
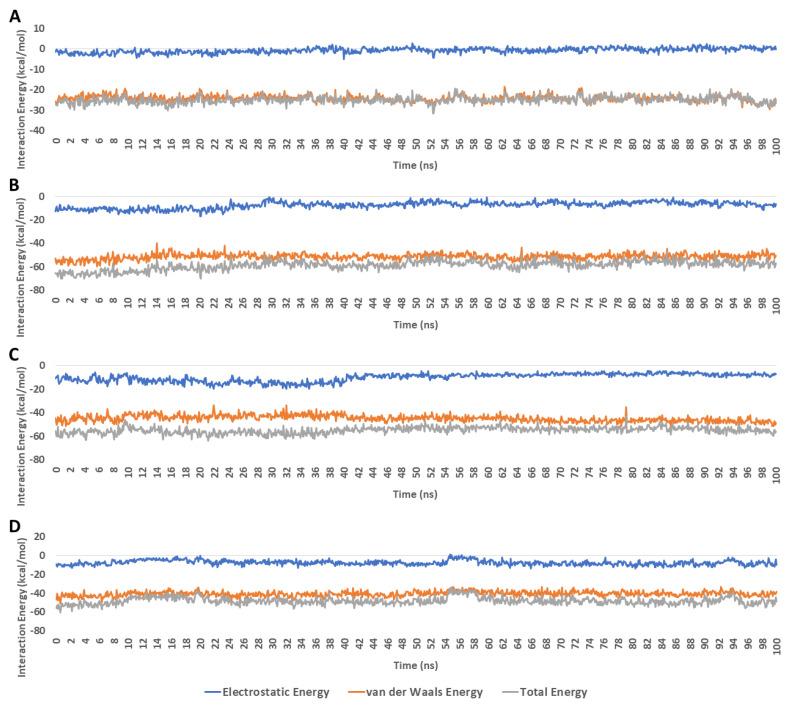
Interaction energies of calycopteretin-3-rutinoside, withanoside IX, and 4,16-dihydroxy-5*β*,6*β*-epoxyphysagulin D along with the co-crystallized inhibitor inside the active site of NMT throughout 100 ns MDS runs (**A**–**D**), respectively). The total interaction energy is the sum of both electrostatic and van der Waals energies.

**Table 1 metabolites-13-00093-t001:** Standard antileishmanial chemotherapeutic drugs and their suggested mode of antileishmanial effect.

Drug		Mode of Action
** 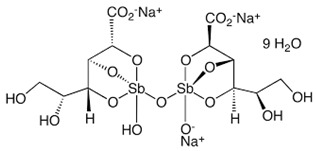 ** **Sodium stibogluconate**	** 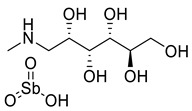 ** **Meglumine antimoniate**	Inhibit the parasite’s glycolysis and fatty acids *β*-oxidation
**Amphotericin B** ** 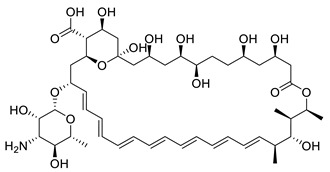 **		Binds the membrane sterols of the parasite and alters its permeability to K^+^ and Mg^2+^ selectively
**Pentamidine** ** 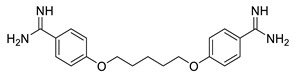 **		Interferes with DNA synthesis and modifies the morphology of the kinetoplast
**Miltefosine** ** 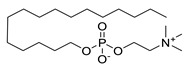 **		Associated with leishmanial alkyl-lipid metabolism and phospholipid biosynthesis
**Paromomycin** ** 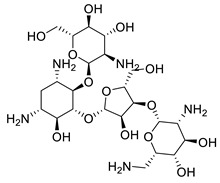 **		Inhibits the protein biosynthesis in sensitive *Leishmania* parasites

**Table 2 metabolites-13-00093-t002:** List of the compounds isolated from *W. somnifera* along with their molecular docking results.

No.	Name	Pose Score (kcal/mol)	Reference for Compound Isolation
**Alkaloids**			
**1**	Somniferine	−16.4	[43]
**2**	D-*α*-Aminoadipic acid	−13.8	[44]
**3**	Anaferine	−12.6	[45]
**4**	Anahygrine	−11.2	[45]
**5**	Tropine	−11.2	[46]
**6**	Cuscohygrine	−10.7	[45]
**7**	Isopelletierine	−10.4	[47]
**8**	Putrescine	−10.0	[48]
**9**	*γ*-Aminobutyric acid	−9.6	[44]
**10**	Withasomnine	−9.0	[49]
**Phenolic compounds**			
**1**	Calycopteretin-3-rutinoside *	−23.3	[50]
**2**	N-*trans*-feruloyl-methoxytyramine	−16.9	[51]
**3**	Naringenin	−15.8	[44]
**4**	Quercetin	−15.7	[44]
**5**	Kaempferol	−15.4	[44]
**6**	Catechin	−14.4	[44]
**7**	Withaninsam A	−13.5	[52]
**8**	Butein	−13.1	[53]
**9**	Withaninsam B	−12.6	[52]
**10**	Vanillic acid	−12.4	[54]
**11**	Syringic acid	−12.2	[55]
**12**	Acetosyringone	−12.0	[56]
**13**	Aesculetin	−11.7	[57]
**14**	Podocarpic acid	−10.3	[44]
**15**	*P*-Coumaric acid	−9.1	[58]
**Sterols**			
**1**	3*β*-Stigmasta-5,24-dien-3-ol	−13.6	[59]
**2**	Campesterol	−13.6	[44]
**3**	Stigmasterol acetate	−13.5	[60]
**4**	*β*-Sitosterol oleate	−13.3	[60]
**5**	*β*-Sitosterol	−11.7	[61]
**6**	Cholesterol	−11.6	[44]
**7**	Stigmasterol	−11.3	[61]
**8**	Brassicasterol	−11.1	[44]
**9**	Crinosterol	−11.0	[60]
**10**	3*β*-Ergosta-5,24-dien-3-ol	−10.5	[59]
**Withanones**			
**1**	Isowithanone	−15.6	[62]
**2**	27-Hydroxywithanone	−15.1	[63]
**3**	4*α*-Hydroxywithanone	−15.1	[57]
**4**	2,3-Dihydro-3*β*-hydroxywithanone	−15.0	[64]
**5**	Withanone	−14.1	[65]
**6**	14*β*-Hydroxywithanone	−13.8	[66]
**Chloride containing withanolides**			
**1**	Withanolide C	−15.7	[67]
**2**	4-Deoxyphysalolactone	−14.1	[67]
**3**	(4*β*,5*β*,6*α*,22R) 5-Chloro-4,6,27-trihydroxy-1-oxowitha-2,24-dienolide 27-Acetate	−12.1	[68]
**4**	Withanolide D chlorohydrin	−12.1	[69]
**5**	6*α*-Chloro-5*β*,17*α*-dihydroxywithaferin A	−11.8	[70]
**6**	Withanolide Z	−11.7	[28]
**Sulfur-containing withanolides**			
**1**	Withanolide sulfoxide	−17.7	[71]
**2**	5*α*,17*α*-Dihydroxy-6*α*,7*α*-epoxy-1-oxo-3*β*-*O*-sulfate-witha-24-enolide	−15.4	[72]
**3**	2,3-Dihydrowithanone-3*β*-*O*-sulfate	−14.6	[73]
**4**	2,3-Dihydrowithaferin A-3*β*-*O*-sulfate	−14.2	[73]
**5**	Ashwagandhanolide	−14.2	[74]
**Withanamides**			
**1**	Withanamide F *	−18.4	[75]
**2**	Withanamide H	−17.3	[75]
**3**	Withanamide E	−16.5	[75]
**4**	Withanamide C	−15.7	[75]
**5**	Withanamide G	−15.5	[75]
**6**	Withanamide B	−15.4	[75]
**7**	Withanamide A	−15.3	[75]
**8**	Withanamide D	−15.0	[75]
**9**	Withanamide I	−14.2	[75]
**Withanolides**			
**1**	Withanolide A *	−18.7	[66]
**2**	(4*β*,5*β*,6*β*,17*α*,22*R*) 5,6-Epoxy-4,17,27-trihydroxy-1-oxowitha-2,24-dienolide *	−18.5	[66]
**3**	Somniferanolide	−18.0	[76]
**4**	Withanolide H	−17.7	[77]
**5**	17-Isowithanolide E	−17.6	[78]
**6**	Withanolide K	−17.4	[79]
**7**	(20*R*,22*R*)14,20-Dihydroxy-1-oxowitha-2,4,6,24-tetraenolide	−17.4	[66]
**8**	Withacoagulin I	−17.3	[44]
**9**	3*α*-(Uracil-1-yl)-2,3-dihydrowithaferin A	−17.1	[80]
**10**	14,17-Dihydroxywithanolide R	−17.0	[57]
**11**	27-Hydroxywithanolide D	−16.6	[81]
**12**	Withanolide D	−16.6	[82]
**13**	24,25-Dihydro-27-desoxywithaferin A	−16.4	[83]
**14**	Somniwithanolide	−16.3	[76]
**15**	Withanolide S	−16.1	[82]
**16**	5,6:14,15-Diepoxy-4,27-dihydroxy-1-oxowitha-2,24-dienolide	−16.0	[68]
**17**	(3*α*,4*β*,5*β*,6*α*,22*R*) 3,6-Epoxy-4,5,27-trihydroxy-1-oxowith-24-enolide	−15.9	[84]
**18**	3*β*-(Uracil-1-yl)-2,3-dihydrowithaferin A	−15.7	[80]
**19**	Tubocapsanolide F	−15.6	[85]
**20**	4-Hydroxywithanolide E	−15.6	[86]
**21**	Withanolide E	−15.4	[79]
**22**	(3*β*,5*α*,6*α*,7*α*,17*α*,22R) 6,7-Epoxy-3,5,17-trihydroxy-1-oxowith-24-enolide	−15.4	[72]
**23**	Quresimine B	−15.3	[87]
**24**	Sominolide	−15.3	[88]
**25**	Withanolide Ws 1	−15.3	[89]
**26**	Withanolide L	−15.2	[79]
**27**	Withasomniferol C	−15.2	[90]
**28**	Withacoagin	−15.2	[91]
**29**	Somniferawithanolide	−15. 2	[76]
**30**	6,7-Epoxy-5,23-dihydroxy-1-oxowitha-2,24-Dienolide	−15.0	[92]
**31**	Withaoxylactone	−14.8	[93]
**32**	Dihydrowithaferin A	−14.8	[63]
**33**	Withanolide J	−14.7	[79]
**34**	5-Deoxywithanolide R	−14.7	[94]
**35**	Withanolide I	−14.7	[79]
**36**	17*α*-Hydroxywithanolide D	−14.5	[81]
**37**	Withasomnilide	−14.4	[76]
**38**	(3*β*,5*α*,6*α*,7*α*,20R,22R) 6,7-Epoxy-3,5,20-trihydroxy-1-oxowith-24-enolide	−14.4	[95]
**39**	Withanolide G	−14.3	[79]
**40**	3*β*-*O*-Butyl-2,3-dihydrowithaferin A	−14. 3	[80]
**41**	27-Deoxywithaferin A	−14.3	[96]
**42**	Pubesenolide (sominone)	−14.3	[97]
**43**	Quresimine A	−14.2	[93]
**44**	27-Hydroxywithanolide B	−14.2	[28]
**45**	Withanolide B	−14.2	[98]
**46**	Withasomniferanolide	−14.1	[76]
**47**	Somnifericin	−14.1	[99]
**48**	20-Deoxywithanolide D	−14.1	[100]
**49**	(5*α*,6*α*,7*α*,16*β*,17(20)E) 6,7-Epoxy-5,16-dihydroxy-1-oxowitha-2,17(20),24-trienolide 16-acetate	−14.1	[98]
**50**	Withanolide M	−14.0	[79]
**51**	Withanolide O	−14.0	[101]
**52**	Dunawithagenin	−13.9	[102]
**53**	5,6-Epoxy-4-hydroxy-1-oxowitha-2,16,24-trienolide	−13.9	[72]
**54**	Withanolide U	−13.9	[101]
**55**	(5*α*,17*α*OH,22*R*) 5,17-Dihydroxy-1-oxowitha-2,6,24-trienolide	−13.7	[103]
**56**	4-Deoxywithaperuvin	−13.7	[104]
**57**	5,6-Epoxy-20-hydroxy-1,4-dioxowitha-2,24-dienolide	−13.7	[105]
**58**	17-Hydroxywithaferin A	−13.6	[106]
**59**	2,3-Dehydrosomnifericin	−13.5	[107]
**60**	5,6-Epoxy-4-hydroxy-1-oxowitha-2,14,24-trienolide	−13.5	[108]
**61**	Withanolide Q	−13.5	[92]
**62**	Withasomniferin A	−13.5	[94]
**63**	(14*α*,20*R*,22*R*)14,20-Dihydroxy-1-oxowitha-2,5,16,24-tetraenolide	−13.5	[109]
**64**	3*β*-(Adenin-9-yl)-2,3-dihydrowithaferin A	−13.4	[80]
**65**	Withanolide F	−13.4	[79]
**66**	Withanolide T	−13.3	[66]
**67**	Withanolide Y	−13.3	[110]
**68**	Withanolide N	−13.3	[82]
**69**	Withacoagulin G	−13.2	[44]
**70**	(4*β*,5*β*,6*β*,20*R*,22*R*) 5,6-Epoxy-4,20-dihydroxy-1-oxowith-24-enolide	−13.2	[105]
**71**	14*α*-Hydroxywithanolide D	−13.2	[111]
**72**	4-Dimethyloxocyclopropyl-2,3-dihydrowithaferin A	−13.1	[83]
**73**	24,25-Dihydrowithanolide D	−13.1	[71]
**74**	Withanolide P	−12.8	[81]
**75**	5,6-Epoxy-20-hydroxy-1,4-dioxowith-2-Enolide	−12.6	[112]
**76**	Withasomniferol B	−12.5	[113]
**77**	Withaferin A	−12.5	[114]
**78**	Withasomniferol A	−12.5	[113]
**79**	5,6-Epoxy-4,20-Dihydroxy-3-methoxy-1-oxowithanolide	−12.2	[115]
**80**	27-Deoxy-14-hydroxywithaferin A	−12.2	[116]
**81**	Withanolide R	−11.9	[92]
**82**	27-Hydroxywithanolide I	−11.7	[102]
**83**	Viscosalactone B	−11.6	[117]
**84**	5-Ethoxy-6,14,17,20-tetrahydroxy-1-oxowitha-2,24-dienolide	−11.6	[102]
**85**	Withalactone	−11.5	[93]
**86**	(1*α*,3*β*,5*α*,6*α*,7*α*,20S,22*R*) 6,7-Epoxy-1,3,5-trihydroxywith-24-enolide	−11.3	[118]
**87**	Withasomidienone	−10.1	[119]
**Withanosides**			
**1**	4,16-Dihydroxy-5*β*,6*β*-epoxyphysagulin D *	−24.0	[83]
**2**	Withanoside IX *	−22.2	[84]
**3**	Physagulin D (1→6)-*β*-D-glucopyranosyl-(1→4)-*β*-D-glucopyranoside *	−22.1	[83]
**4**	Withanoside VIII	−20.5	[84]
**5**	Withanoside X *	−19.9	[84]
**6**	Withanoside II *	−19.6	[120]
**7**	Withanoside IV	−19.2	[120]
**8**	24,25-Dihydrowithanoside VI *	−18.5	[75]
**9**	Sitoindoside IX *	−18.1	[121]
**10**	Withanoside III	−17.8	[120]
**11**	Withanoside VII	−17.5	[120]
**12**	Withanoside V	−16.6	[120]
**13**	Withanoside VI	−16.3	[120]
**14**	Glucosomniferanolide	−16.1	[90]
**15**	Withanoside XI	−15.7	[84]
**16**	Sitoindoside VII	−14.4	[122]
**17**	Sitoindoside VIII	−14.0	[122]
**18**	Withanoside I	−14.0	[120]
**19**	Sitoindoside X	−12.9	[121]

* Compounds with top-scoring docking results.

**Table 3 metabolites-13-00093-t003:** Binding free energies (MM-PBSA) of top-scoring compounds along with the co-crystalized inhibitor in complex with NMT.

Energy Component	Calycopteretin 3-Rutinoside	Withanoside IX	4,16-Dihydroxy-5,6- epoxyphysagulin D	Co-Crystalized Inhibitor
Δ*G*_gas_	−17.98	−24.63	−21.45	−28.73
Δ*G*_solv_	9.76	16.15	11.64	15.44
Δ*G*_Total_	−8.21	−8.47	−9.80	−13.29

## Data Availability

The data, which include the 2D and 3D structures of the compounds under investigation, are obtainable from the corresponding author upon request.

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
