# Peer review of "Identification of Potential Leishmania N-Myristoyltransferase Inhibitors from Withania somnifera (L.) Dunal: A Molecular Docking and Molecular Dynamics Investigation"

_metabolites, 2023, doi:10.3390/metabo13010093_

Round 1
Reviewer 1 Report
Orabi et al. studied the effect of Withania somnifera against NMT drug target using computational methods. After reading the manuscript I have found that there are many things that need to be improved before consideration.
In the introduction, please provide the rationale behind the selection of NMT as a drug target and provide the background of known inhibitors.
The 3D structure representation is very poor it should be improved.
Results sections lack a detailed explanation of the results.
How the MD simulation time was decided. Authors should extend MD simulations to 100 ns.
Moreover, results should be presented in detail for RMSD/RMSF/H-bonds/Binding free energy
How the hit candidates were selected. In Table 3 selected compounds displayed poor binding affinity than known inhibitor?
Author Response
Our responses to the reviewer's comments are shown in the attached file.

Reviewer 2 Report
The methodology of the manuscript is satisfactorily presented and well delimited. The results are solid and well discussed. I have only minor comments: Please, in the introduction, justify the choice of evaluation of N-myristoyltransferas over other targets. It would be interesting for the authors to add other targets of interest already investigated for leishmania. If NMT has already been extensively investigated, make it clear how this study stands out from the rest.
Author Response
Our responses to the reviewer's comments are shown in the attached file

Round 2
Reviewer 1 Report
I still recommend 100 NS MDS studies if the system displayed stable fluctuations it does not mean ligands will stay in the same phase with the increase of time there might be a chance to get several conformations. Moreover, complex bound with Calyopteretin not seems stable when compared to others how authors will justify this?
Author Response
Our responses are shown in the attached file

Round 3
Reviewer 1 Report
The authors revised their manuscript according to my suggestions. Therefore study can be considered for publication. Moreover, a few minor changes can be considered for further improvement in this paper.
1) In Figure 5-7 please remove the yellow color and add proper labels to the residues and show hydrogen bonds.
2) Figure 8-9, on the X-access, please use normal numbers instead of decimals similar to Figure 10
3) Figure 3. Can also be improved using proper labeling.
Author Response
We have responded to the reviewer's comment as shown by the track changes function of the word. A response letter is shown in the attachment.
